# Pruritus Is Associated with an Increased Risk for the Diagnosis of Autoimmune Skin Blistering Diseases: A Propensity-Matched Global Study

**DOI:** 10.3390/biom13030485

**Published:** 2023-03-06

**Authors:** Ulrike Raap, Maren M. Limberg, Khalaf Kridin, Ralf J. Ludwig

**Affiliations:** 1Clinics of Dermatology and Allergy, Division of Experimental Allergy and Immunodermatology, University of Oldenburg, 26129 Oldenburg, Germany; 2Lübeck Institute for Experimental Dermatology, University of Lübeck, 23560 Lübeck, Germany; 3Azrieli Faculty of Medicine, Bar-Ilan University, Safed 5290002, Israel; 4Unit of Dermatology and Skin Research Laboratory, Barch Padeh Medical Center, Poriya 15208, Israel; 5Department of Dermatology, University Clinic of Schleswig-Holstein, 23560 Lübeck, Germany

**Keywords:** pruritus, pemphigus, pemphigoid, case–control study

## Abstract

Autoimmune bullous skin diseases (AIBDs), such as bullous pemphigoid (BP) and pemphigus, are characterized and caused by autoantibodies targeting structural proteins. In BP, clinical experience and recent systematic evaluation identified pruritus to be common and an important cause of impaired quality of life. Furthermore, chronic pruritus may be the sole clinical symptom of BP. In pemphigus, a retrospective study recently documented a high prevalence of pruritus. The temporal relation between pruritus and BP/pemphigus are, however, unknown. Likewise, the presence of pruritus in AIBDs other than BP and pemphigus is unknown. To address this, we performed propensity-matched retrospective cohort studies using TriNetX, providing real-world patient data to (i) assess the risk to develop AIBDs following the diagnosis of pruritus and (ii) vice versa. We assessed this in eight AIBDs: BP, mucous membrane pemphigoid (MMP), epidermolysis bullosa acquisita, dermatitis herpetiformis, lichen planus pemphigoides (LPP), pemphigus vulgaris, pemphigus foliaceous, and paraneoplastic pemphigus (PNP). For all AIBDs, pruritus was associated with an increased risk for the subsequent diagnosis of each of the eight investigated AIBDs in 1,717,744 cases (pruritus) compared with 1,717,744 controls. The observed hazard ratios ranged from 4.2 (CI 3.2–5.5; *p* < 0.0001) in MMP to 28.7 (CI 3.9–211.3; *p* < 0.0001) in LPP. Results were confirmed in two subgroup analyses. When restricting the observation time to 6 months after pruritus onset, most HRs noticeably increased, e.g., from 6.9 (CI 6.2–7.9; *p* < 0.0001) to 23.3 (CI 17.0–31.8; *p* < 0.0001) in BP. Moreover, pruritus frequently developed following the diagnosis of any of the eight AIBDs, except for PNP. Thus, all AIBDs should be considered as differential diagnosis in patients with chronic pruritus.

## 1. Introduction

Autoimmune bullous skin diseases (AIBDs) are characterized and caused by autoantibodies targeting structural proteins [1]. Depending on the clinical presentation and the targeted autoantigen(s), 12 different AIBDs can be diagnosed [2,3,4]. Bullous pemphigoid (BP), pemphigus vulgaris (PF), and pemphigus foliaceous (PF) are the most common AIBDs, although all are still considered orphan diseases [5,6]. Despite their low prevalence, AIBDs impose a significant medical burden, for both patients and the healthcare system. This is mainly due to the limited and partially insufficient treatment options, as well as a significant, often treatment-associated, (co)-morbidity [7,8]. 

Pruritus represents one of the most bothersome symptoms in patients and has a strong impact on quality of life. Pruritus is defined as chronic if it persists more than 6 weeks [9]. Interestingly, pruritus not only occurs in allergic skin diseases, such as atopic dermatitis and contact eczema, but also in chronic urticaria, lichen planus, psoriasis, and AIBD—most prominently in BP. This indicates different immunological pathways for the orchestration of pruritus. Clinically, pruritus leads to an intense desire to scratch the skin in atopic dermatitis [10]. Strikingly and by contrast, this is not the case for other diseases, e.g., in chronic urticaria. Thus, the underlying pathophysiology of pruritus is yet not fully understood. Free nerve endings of nonmyelinated C-type nerve fibers that are located at the dermo-epidermal junction and within the epidermis are assumed to regulate pruritus [11]. Further, there is also evidence of histamine-independent itch-specific fibers in the skin, such as G-protein-coupled receptors, kappa-opioid receptors, and mu-opioid receptors [12]. More recently investigated receptors include the interleukin (IL)-31RA and oncostatin-m-specific receptor (OSMR), in addition to the transient receptor potential cation channel subfamily V member 1 (TRPV1) and transient receptor potential ankyrin 1 (TRPA1), thymic stromal lymphopoietin protein receptor (TSLPR), protease-activated receptor 2 (PAR-2), neurokinin 1 receptor (NK1R), Trk, and histamine (H) 1R and H4R, as well as mas-related G-protein-coupled receptors (MRGPRs) [13,14,15,16,17] that play a role in pruritus mechanisms (Figure 1). Regarding the inflammatory infiltration and possible neuroimmune-interacting mechanisms, it has recently become evident that resident cells, such as mast cells, keratinocytes, and sensory neurons, as well as transient infiltrating cells, such as T cells, basophils, and eosinophils, play an important role in the activation of peripheral nerves and neuroimmune interaction mechanisms of pruritus [18]. In this regard, it has been shown that neuronal sprouting is induced by mediators, including IL-31, and nerve growth factors, including brain-derived neurotrophic factor, which have been also detected in BP skin [19,20,21].

Based on clinical experience, pruritus has long been considered a cardinal symptom of BP [3]. This notion has recently been validated in a multicenter, prospective observational study [22]: Here, a total of 60 BP patients was recruited. Of these, 85% reported the daily occurrence of pruritus, which caused a significant impact on quality of life. Of note, in some patients, pruritus is the sole clinical presentation of BP [23,24]. Hence, BP should be considered as a differential diagnosis in (elderly) patients presenting with chronic pruritus. In the two common forms of pemphigus, pemphigus vulgaris (PV) and pemphigus foliaceus (PF), pruritus had, until recently, only been mentioned in anecdotal clinical observations—with the exception of four, retrospective, single-center analyses. In the first study stemming from Iran, 61 pemphigus patients were enrolled. The most common complaint was a burning sensation, which was observed in over 80% of the patients. Itch was experienced by 47.5% of the patients [25]. In a recent study from the US, 38 PV and 20 PF patients were recruited to investigate the presence of pruritus [26]. Overall, more than 80% of pemphigus patients reported the presence of pruritus, which was comparable to BP patients, which were recruited as controls. Comparing PV and PF, pruritus was more severe in the latter. Hence, pruritus should be evaluated and, if needed, addressed in pemphigus patients. Data from Poland, as well as an independent study from the US for BP, pemphigus, and MMP, showed similar results in BP patients [27,28]. For rarer AIBDs, insights on epidemiology, including the presence of pruritus, are even more sparse, or absent. 

To address the potential and temporal association between AIBD and pruritus, we used TriNetX, providing access to electronic health records from over 100 million patients. We performed propensity-matched studies, in which we first addressed if pruritus precedes the diagnosis of any of the eight selected AIBDs, namely BP, mucous membrane pemphigoid (MMP), epidermolysis bullosa acquisita (EBA), dermatitis herpetiformis (DH), lichen planus pemphigoides (LPP), PV, PF, and paraneoplastic pemphigus (PNP). We performed this analysis without restriction to the follow-up, and repeated the same study, but restricted the follow-up to 6 months. Next, we investigated the risk of pruritus following the diagnosis of any of the researched AIBDs.

Foremost, insights into the temporal relationship between pruritus and AIBDs will be essential for patient management. More specifically, if AIBDs are considered as a differential diagnosis in pruritus patients, early diagnosis may be facilitated. In this regard, it has also been shown that subtypes of BP, including non-bullous BP, are associated with heterogenous skin lesions predominantly featuring pruritus, indicating that this symptom is of major importance as a possible key symptom for BP [23]. Furthermore, targeting pruritus in established AIBDs could improve quality of life. Often the treatment of pruritus is challenging since H1-antihistamines are not effective. This is explained by the fact that several AIBDs are associated with granulocytic infiltration, including basophils, which are the prime early producers of IL-4 and 13. In this regard, it has been shown recently that targeting IL-13 and IL-4 using the monoclonal antibody dupilumab resolves pruritus and inflammatory skin lesions in BP, indicating a role of neuroimmune interaction mechanisms with immune cells [29,30].

## 2. Materials and Methods

### 2.1. Study Design and Database

The Global Collaborative Network of TriNetX was used for all studies. As part of a collaboration between the University Clinic of Schleswig-Holstein (UKSH) and TriNetX, UKSH researchers had free access to the TriNetX network.

### 2.2. Study Population and Definition of Eligible Patients

The data used in this study were collected between 10th and 11th October 2022, from the TriNetX Global Collaborative Network, which provided access to electronic medical records (diagnoses, procedures, medications, laboratory values, genomic information) from approximately 114 million patients from 91 healthcare organizations (HCO). The data mainly stem from US HCOs. Specifically, 96% of the data are derived from this region, while 4% of the data are from HCOs outside the USA. We here first compared the risk to develop any AIBD (outcome definitions displayed in Table 1, Table 2, Table 3 and Table 4) between patients diagnosed with pruritus (ICD10CM:L29) to those without (ICD10CM:Z00 not ICD10CM:L29). This analysis was performed without any time restrictions for follow-up and with follow-up limited to the first 6 months after the diagnosis of pruritus. Next, we assessed the risk of subsequent pruritus development (outcome defined as ICD10CM:L29) in patients with AIBD. Each AIBD cohort was build using the respective ICD10CM code (Table 5), excluding those with an ICD10CM code for other AIBDs. Individuals without any AIBD (defined by ICD10CM:Z00 not ICD10CM:L10 not ICD10CM:L12 not ICD10CM:L43.1 not ICD10CM:L13.0) were used as controls. Propensity score matching was performed for the following variables: age, sex, ethnicity, and skin color. In all analyses, outcomes prior to the respective index event were excluded. 

To exclude the presence of cholestasis or uremia confounds the results; therefore, we performed a subgroup analysis where we excluded cases and controls (for all groups) that had elevated bilirubin concentrations in serum, plasma, or blood (TNX:9050, most recent laboratory, >1.2 mg/dL), or elevated urea nitrogen in serum, plasma, or blood (TNX:9030, most recent laboratory, ≥24 mg/dL). To exclude the presence of other inflammatory skin diseases in which pruritus is a common symptom is a confounder; therefore, we performed a subgroup analysis comparing cases (pruritus, ICD10:L29) to controls (ICD10:Z00) excluding (for both groups) ICD10 codes for urticaria (ICD10:L50), psoriasis (ICD10:L40) and dermatitis and eczema (ICD10:L20-L28 and L30) present at any until the index event (ICD10:L29 or ICD10:Z00). Both subgroup analyses were performed on the Global Collaborative Network on 25 January 2023.

TriNetX, LLC is compliant with the Health Insurance Portability and Accountability Act (HIPAA), the US federal law which protects the privacy and security of healthcare data and any additional data privacy regulations applicable to the contributing HCO. TriNetX is certified to the ISO 27001:2013 standard and maintains an Information Security Management System (ISMS) to ensure the protection of the healthcare data it has access to, meeting the requirements of the HIPAA Security Rule. Any data displayed on the TriNetX Platform in aggregate form, or any patient-level data provided in a data set generated by the TriNetX Platform, only contain de-identified data as per the de-identification standard defined in Section §164,514(a) of the HIPAA Privacy Rule. The process by which the data are de-identified is attested to through a formal determination by a qualified expert as defined in Section §164,514(b)(1) of the HIPAA Privacy Rule. Because this study used only de-identified patient records and did not involve the collection, use, or transmittal of individually identifiable data, this study was exempted from Institutional Review Board approval.

### 2.3. Statistical Analysis

Baseline characteristics were described by means and standard deviations (SDs) for continuous variables and numbers and percentages for dichotomous variables. Continuous variables were compared using the Student t-test. Dichotomous variables were compared by Pearson chi-square test. Survival analyses were conducted by the Kaplan-Meier method. A log-rank test was run to determine if there were differences in the survival distribution for patients in the two investigated groups. Hazard ratios (HR)s for the study outcomes were obtained using the Cox regression model. Nelson–Aalen plots were utilized to test the proportional hazards assumption; two-tailed *p*-values less than 0.05 were considered statistically significant.

## 3. Results

### 3.1. Description of Cohorts

#### 3.1.1. Patients with Pruritus

We recruited 1,720,045 cases with pruritus and a similar number of controls. Cases were 43.3 ± 23.8 years old, 66% were female, 70% were not Hispanic or Latino, and 64% were White. No difference among cases and controls was noted for age, sex, ethnicity, or skin color (Table 1).

#### 3.1.2. Patients with AIBD

On average, 4993 ± 5252 patients were recruited for each AIBD. In more detail, 16,019 cases/controls were included for BP, 4165 for MMP, 1234 for EBA, 7791 for DH, 158 for LPP, 6243 for PV, 4161 for PF, and 176 for PNP. For each AIBD, a similar number of propensity-matched (age, sex, ethnicity, and skin color) controls was included. Similar to the cohort of pruritus patients, no differences regarding epidemiological characteristics were noted between AIBD patients and the corresponding controls (Table 2).

### 3.2. Pruritus Is Associated with A Higher Risk for Subsequent AIBD Diagnosis

#### 3.2.1. Pruritus Is Associated with a Higher Risk for Subsequent AIBD Diagnosis

In our cohort, pruritus was associated with an increased risk for subsequent AIBD diagnosis (Table 3, Figure 2a). Among the AIBDs, the highest proportion of pruritus patients to develop an AIBD was observed for BP. Here, 0.12% of patients with pruritus subsequently developed BP, as opposed to 0.02% in controls. The hazard ratio (HR) for BP development in pruritus amounted to 6.952 (confidence interval (CI) 6.158–7.85; *p* < 0.0001). Pruritus was also associated with an increased risk for all other AIBDs included in our study, notably for MMP (HR 4.219, CI 3.213–5.54; *p* < 0.0001), which primarily manifests at the oral mucosa and only rarely at the skin [31]. The highest degree of association to develop subsequent AIBD in pruritus was found for LPP 28.705 (CI 3.9–211.299; *p* < 0.0001). In more detail, the risk of subsequent EBA following the diagnosis of pruritus was increased (HR 5.402, CI 3.76–7.76; *p* < 0.0001), as well as for DH (HR 6.068, 5.09–7.234; *p* < 0.0001), LPP (HR 28.705, CI 3.9–211.299; *p* < 0.0001), PV (4.582, CI 3.754–5.592; *p* < 0.0001), PF (HR 5.093, CI 3.938–6.587; *p* < 0.0001), and PNP (HR 6.913, CI 2.703–17.678); *p* < 0.0001).

To exclude confounding by the presence of cholestasis or uremia, subgroup analysis was performed for the three most prevalent AIBDs within the TriNetX network (BP, DH, and PV), excluding cases and controls with elevated bilirubin or urea nitrogen concentrations in serum, plasma, or blood. As expected, cohort sizes decreased. As for the cohorts prior to the consideration of these laboratory results, no significant differences were noted among the factors after propensity matching (Table 1). In this subgroup analysis, the risk for developing BP, DH, or PV following pruritus was comparable to the analysis that did not exclude elevated bilirubin or urea nitrogen concentrations (Table 3). In more detail, the HRs in the subgroup analysis were 6.794 (CI 5.927–7.788; *p* < 0.0001), 5.545 (CI 4.662–6.596; *p* < 0.0001), and 4.749 (CI 3.775–5.976; *p* < 0.0001), for BP, DH, and PV, respectively. This indicates that the development of BP, DH, and PV following pruritus is mainly independent of cholestasis or uremia.

To exclude the presence of other inflammatory skin diseases in which pruritus is a common symptom is a confounder; therefore, we performed a subgroup analysis excluding ICD10CM codes for inflammatory skin diseases in which pruritus is commonly observed from both groups. The cohorts showed no difference regarding age, sex, ethnicity, or skin color (Table 1). The risk of developing BP, DH, or PV following the diagnosis of pruritus was also significantly increased when the aforementioned inflammatory skin diseases were excluded. However, compared with the initial analysis and the subgroup analysis considering elevated bilirubin or urea nitrogen concentrations, the hazard rations were lower (Table 3). In more detail, the HRs in the subgroup analysis, stratified to exclude the presence of other inflammatory skin diseases, were 4.878 (CI 4.11–5.789; *p* < 0.0001), 3.509 (CI 2.753–4.472; *p* < 0.0001), and 3.857 (CI 2.938–5.063; *p* < 0.0001), for BP, DH, and PV, respectively. This indicates that pruritus is associated with an increased risk for BP, DH, and PV, independent of the presence of other inflammatory skin diseases. 

#### 3.2.2. The Majority of AIBDs Are Diagnosed within 6 Months after the Initial Pruritus Manifestation

To understand the temporal relation between pruritus and AIBD, we limited the follow-up to 6 months after the diagnosis of pruritus. Again, pruritus was identified to be associated with an increased risk for subsequent AIBD diagnosis (Table 4, Figure 2b). For all AIBDs, the risk for AIBD diagnosis following pruritus manifestation profoundly increased when limiting the follow-up to 6 months. As such, the majority of AIBD cases were diagnosed within the first 6 months following pruritus (Table 4 and Table 5, Figure 2). In more detail, 2003 BP cases were diagnosed in 1,717,744 patients with pruritus when the time for follow-up was not limited. In the analysis limited to 6 months following the pruritus diagnosis, 973 subsequent BP cases were noted in the same cohort of patients with pruritus. Furthermore, compared with the analysis with no limitation to follow-up time, the HR increased considerably from 6.952 to 23.293 (CI 17.042–31.838; *p* < 0.0001) when analysis was restricted for 6 months following the diagnosis of BP. 

Interestingly, similar findings were also made for most other AIBDs: Regarding mucous membrane pemphigoid, 256 cases were identified in patients with pruritus in the non-time restricted follow-up analysis. Again, in the same cohort, 79 were diagnosed within 6 months after the diagnosis of pruritus was made. This is also reflected by the further increased risk for the development of MMP in patients with pruritus in the time-restricted follow-up (HR 6.459, CI 3.519, 11.855; *p* < 0.0001). Along the same lines, a total of 180 EBA cases occurred throughout the entire follow-up. Of these, 65 occurred within the first 6 months. This amounted to a HR of 12.756 (CI 5.137–31.679; *p* < 0.0001) in the analysis where follow-up was limited to 6 months. For DH, 341 of 843 cases occurred within 6 months after pruritus diagnosis, again resulting in an even further increased risk of DH development in this cohort (HR 14.535, CI 9.529–22.171; *p* < 0.0001). In LPP, 10/27 cases occurred within the first 6 months. For PV, of the 1,717,578 pruritus cases included, 522 subsequently developed PV at any time point during the follow-up. Of these, 246 manifested within 6 months following pruritus. For PF, 1,717,792 pruritus patients were recruited, and a total of 343 was diagnosed with subsequent PF, of which 179 developed the disease within the first 6 months. Lastly, for PNP, 21 of a total of 34 cases were diagnosed within 6 months following pruritus.

### 3.3. AIBDs Are Associated with an Increased Risk for the Subsequent Diagnosis of Pruritus

Lastly, we addressed if AIBDs are associated with an increased risk for subsequent pruritus manifestation. Here, the probability of pruritus was compared in patients diagnosed with any of the AIBDs investigated with individuals without any AIBD. Apart from PNP, all AIBDs imposed an increased probability for subsequent pruritus manifestation (Table 5). To exclude confounding by the presence of cholestasis or uremia, subgroup analysis was performed for the three most prevalent AIBDs within the TriNetX network (BP, DH, and PV), excluding cases and controls with elevated bilirubin or urea nitrogen concentrations in serum, plasma, or blood. As expected, cohort sizes decreased. As for the cohorts prior to the consideration of these laboratory results, no significant differences were noted among the factors after propensity matching (Table 2). In this subgroup analysis, the risk for developing pruritus following either BP, DH, or PV was comparable to the analysis that did not exclude elevated bilirubin or urea nitrogen concentrations (Table 5). In more detail, the HRs in the subgroup analysis were 5.093 (CI 3.989–6.504; *p* < 0.0001), 5.632 (CI 3.951–8.03; *p* < 0.0001), and 4.043 (CI 2.533–6.454; *p* < 0.0001), for BP, DH, and PV, respectively. This indicates that the development of pruritus in these AIBDs is mainly independent of cholestasis or uremia.

## 4. Discussion

The major finding of our study is the identification of an increased probability for AIBD development following the diagnosis of pruritus. There are two possible explanations for this finding: First, pruritus could promote AIBD development. Alternatively, pruritus could be an early symptom of many AIBDs. Based on the relatively short time between pruritus and AIBD manifestation, as well as shared pathogenic pathways in pruritus and BP, we favor the latter assumption. Recently, it has been shown that the early phase of BP is associated with an accumulation of basophils that correlates with eosinophils [32]. This finding is interesting, given the fact that basophils are a source of IL-31 [33], which is supposed to be one of the main cytokines in pruritus [34]. Indeed, IL-31 not only correlates with pruritus in inflammatory skin diseases, such as atopic dermatitis and urticaria [35,36], there is also a trend between pruritus severity and IL-31 expression in the skin of patients with BP [37]. In the latter study, it had been shown that the majority of IL-31-positive cells had been eosinophils. This finding is confirmed by our data in BP patients, showing that eosinophils in the skin and blister fluid are a major source of IL-31 [20]. Interestingly, eosinophils derived from blister fluids showed strong expression of IL-31 too. Since IL-31 also induces nerve sprouting [19], IL-31 bridges the gap between immune cells and the nervous system, indicating neuroimmune interaction mechanisms that lead to an early priming of pruritus [38]. In addition to IL-31, thymic stromal lymphopoietin (TSLP) is a cytokine described as key mediator in pruritus, which can activate neurons to induce itch [39]. TSLP is involved in the pathogenesis of BP [40]. A recent study identified by RNA sequencing that itch-related mediators and receptors, such as phospholipase, substance p, sodium channels, TRP channels, and different cytokines and chemokines, are differentially expressed in pruritic skin [41]. It has been found that many other factors, including mast cell mediators, neuropeptides, neurotrophins, and proteases have many effects on the skin, mainly participating in the occurrence and development of itching [42]. Although the exact mechanism of itch has not been fully elucidated, it is clear that a complex crosstalk between the stratum corneum, keratinocytes, T cells, eosinophils, basophils, mast cells, and nerve fibers plays an important role in the initiation and maintenance of pruritus [43,44,45].

In vitro stimulation of basophils with IL-31 leads to a strong release of IL-4 and IL-13 [33]. IL-4 in turn upregulates the expression and function of the H4R in eosinophils, indicating an autocrine loop for cellular activation [46]. In this regard, several case reports have shown that using dupilumab, a monoclonal antibody targeting IL-4 and IL-13, successfully resolves BP [30,47], in addition to pemphigus vulgaris [48]. Furthermore, successful treatment options for BP and pemphigus include the use of anti-IgE [49,50]. This is of major interest, since IgE-autoantibodies can be found in the serum and skin of non-bullous and bullous pemphigoid patients, both of which suffer from pruritus [51].

Our findings also have critical clinical implications. In patients with chronic pruritus, AIBD should be among the differential diagnoses initially considered. This will facilitate an earlier diagnosis. Given that early treatment with appropriate drugs, i.e., rituximab in pemphigus [52,53], is beneficial, early diagnosis could contribute to the overall outcome. Furthermore, the presence of pruritus needs to be assessed in all AIBD patients and, if present, be included as a symptom to be treated.

Our study has several limitations to be acknowledged. First, patient electronic health record data may suffer from misdiagnosis and/or miscoding and do not encompass all possible confounding factors. Second, the TriNetX database provides access to medical data from individuals who had medical encounters with healthcare systems. Thus, our analysis does not include patients with low access to healthcare facilities. Third, the coding of a symptom such as pruritus may be biased. More specifically, pruritus may not be coded alongside a disease that is associated with itch, such as BP. In our data set this may be the case, as “only” 3.4% of BP patients developed pruritus. This relatively low number of BP patients with pruritus may also be due to the study design, as outcomes prior to the index event were excluded. Furthermore, the coding of ICD10:L29 will most likely be underrepresented when a definite other diagnosis that is associated with pruritus, e.g., BP, has been established. Fourth, the nature of the study does not allow us to draw conclusions on causality. Thus, any casual relationships need to be established in prospective observational clinical trials. 

Previous studies on the subject almost exclusively focused on BP and pemphigus patients. In BP, pruritus has long been considered a cardinal symptom of the disease [3]. This clinical observation has, however, only recently been clearly documented in a multicenter prospective observational study. Here, 60 PB patients were included. Of note, pruritus occurred daily in 85% of patients, with tingling as well as burning sensations. Overall, pruritus in BP was identified as an important cause of impaired quality of life [22]. In a study from Poland, all of the included 28 BP patients (and 24 with DH) reported the presence of itch. Depending on the disease (BP or DH), 20–25% of the patients reported generalized itch, while 75–80% experienced localized itching [27]. Furthermore, two recent studies from the US confirmed a high prevalence of itch in BP patients [26,28]. Regarding pemphigus, a total of four studies documented the presence of itch and/or pruritus in patients: In 2012 an Iranian study addressed the quality of life in 61 pemphigus patients, which also included questions related to itch. Here, itch was present in almost half of the patients, and presence of itch was associated with a decrease in the quality of life [25]. These findings were confirmed by two recent studies stemming from the US: Dr. Cole and coworkers recruited 36 pemphigus patients. They used the ItchyQoL as a survey instrument to measure itch. Whilst the prevalence of itch is not indicated in the study, the ItchyQoL documented high scores. In addition, they also included MMP patients (*n* = 23) in their study and documented the presence of clinically relevant itch in this patient group [28]. The study by Dr. Rolader and colleagues on 58 pemphigus patients were also included. Of these, 38 were diagnosed with PV, and 20 with PF. Within the cohort of pemphigus patients, over 80% of the patients reported the presence of itch. The severity of the itch, assayed by BPDAI pruritus, was moderate to severe [26]. Whilst all of the above studies clearly documented a presence of pruritus in AIBD (mostly BP and pemphigus) patients, the temporal relationship between pruritus and AIBD has so far not been addressed. Furthermore, with the exception of studies that also included MMP or DH, pruritus has not been investigated in rare AIBDs, such as EBA or LPP. Regarding the temporal association of pruritus and AIBD, a landmark study by Dr. Meijer and colleagues, who newly diagnosed BP in 4/125 nursing home residents prompted by the presence of chronic itch [24], hinted towards the possibility that pruritus may precede BP manifestation. 

## 5. Conclusions

In summary, the presence of pruritus is associated with an increased risk for a subsequent diagnosis of AIBDs. In addition, pruritus is a common symptom in AIBD. Thus, all AIBDs should be considered in patients with chronic pruritus, and management of pruritus should be included among the treatment aims for AIBDs. In BP, the inflammatory early phase is characterized by immune cell infiltration, including eosinophils and basophils, which is associated with pruritus. Since novel therapeutic targets, including anti-IL-4/IL-13 and anti-IgE, have successfully been used, immunological pathways are suggested to play an important role in the mechanism of pruritus in AIBD and may be used to target pruritus and overall disease activity in these patients.

## Figures and Tables

**Figure 1 biomolecules-13-00485-f001:**
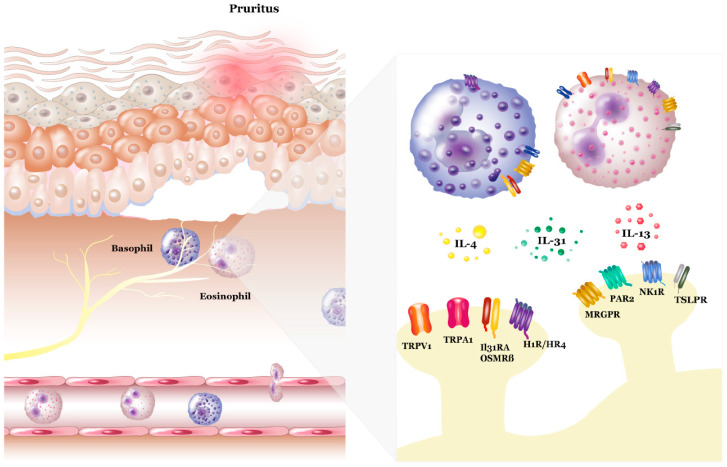
Neuro-immune interaction mechanism in pruritic skin diseases. Crosstalk between keratinocytes, immune cells, and sensory nerves is involved in the pathophysiology of pruritus. Immune cells (eosinophils and basophils) infiltrate the lesional skin from the peripheral blood and release inflammatory and pruritic mediators, such as IL-31, histamine, and BDNF. In addition, neurons modulate the function of immune cells by releasing neurotransmitters and neuropeptides. Several receptors and channels (IL-31 receptor A (IL-31RA), as well as the oncostatin m receptor (OSMR), transient receptor potential vanilloid 1 (TRPV1) and ankyrin 1 (TRPA1), thymic stromal lymphopoietin receptor (TSLPR), protease-activated receptor 2 (PAR-2), neurokinin-1 receptor (NK1R), histamine receptors H1/H4 (H1R/H4R), tropomyosin receptor kinase (Trk), and mas-related G-protein-coupled receptors (MRGPRs), are expressed on neurons and immune cells playing a role in pruritus mechanisms.

**Figure 2 biomolecules-13-00485-f002:**
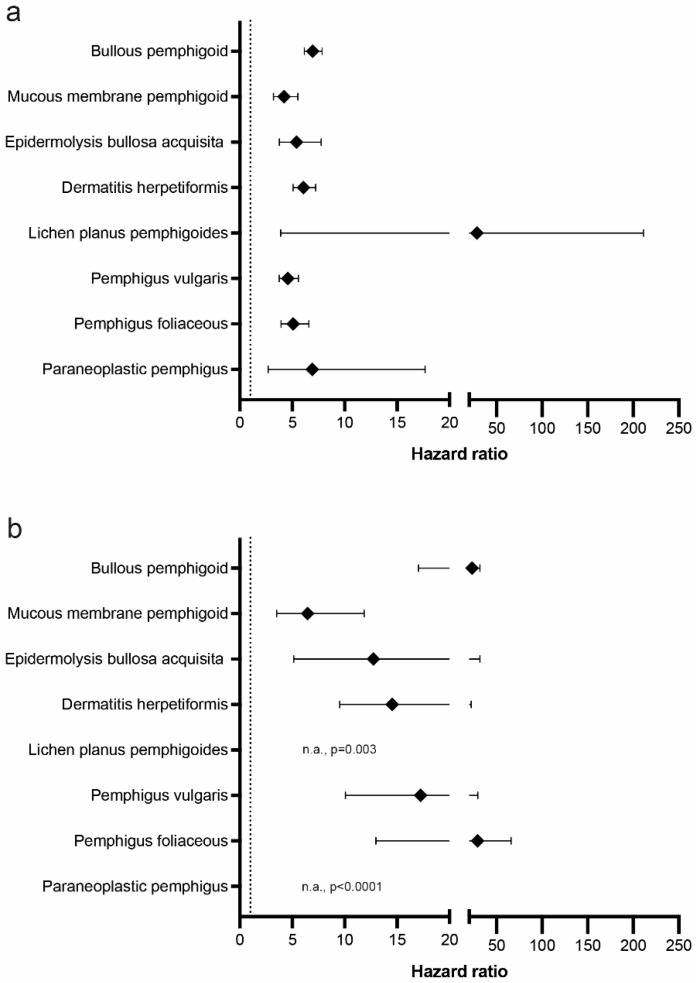
Hazard ratios for autoimmune bullous skin disease development in patients with pruritus. Data from the Global Collaborative Network of TriNetX were used to determine the risk of subsequent development of autoimmune bullous skin diseases (AIBDs). (**a**) Hazard ratio (HR) for development of the indicated AIBDs in patients diagnosed with pruritus compared with those without pruritus. Diamonds indicate HR; error bars indicate 5/95 confidence intervals. (**b**) Same analysis as in panel (**a**) but limiting follow-up time after the diagnosis of pruritus to 6 months. Abbreviations: n.a.: not applicable.

**Table 1 biomolecules-13-00485-t001:** Cohort description for pruritus patients. Abbreviations: SD: standard deviation. * Cases (pruritus) and controls with elevated bilirubin or urea nitrogen concentrations in serum, plasma, or blood were excluded from analysis. ** Cases (pruritus) and controls with any other dermatosis associated with itch were excluded from analysis.

Characteristic	Pruritus	Controls	Pruritus *	Controls *	Pruritus **	Controls **
Number of participants	1,720,045	1,720,045	1,657,855	1,657,855	1,312,591	1,312,591
Age in years (SD)	43.3 ± 23.8	43.3 ± 23.8	42 ± 22.7	42 ± 22.7	43.9 ± 22.3	43.9 ± 22.3
Female (%)	65.869	65.869	67.242	67.242	65.579	65.579
Not Hispanic or Latino (%)	69.503	69.503	74.389	74.389	73.053	73.053
White (%)	64.205	64.205	65.526	65.526	64.81	64.81

**Table 2 biomolecules-13-00485-t002:** Cohort description for pemphigus and pemphigoid diseases. Abbreviations: SD: standard deviation. * Cases and controls with elevated bilirubin or urea nitrogen concentrations in serum, plasma, or blood were excluded from analysis.

Disease	Cases/Controls	Number of Participants	Age in Years (SD)	Female (%)	Not Hispanic or Latino (%)	White (%)
Bullous pemphigoid	Cases	16,019	72.7 ± 14.9	53.543	60.690	64.068
	Controls	16,019	72.7 ± 14.9	53.543	60.678%	64.068%
Bullous pemphigoid *	Cases	13,398	71.8 ± 15.7	55.486	56.046%	56.523%
	Controls	13,398	71.8 ± 15.7	55.486	56.046%	56.523%
Mucous membrane pemphigoid	Cases	4165	66.1 ± 14.9	61.345	68.884%	73.661%
	Controls	4165	66.1 ± 14.9	61.345%	68.884%	73.661%
Epidermolysis bullosa acquisita	Cases	1234	56.4 ± 19	52.107%	48.947%	69.287%
	Controls	1234	56.3 ± 19.3	52.107%	48.298%	69.854%
Dermatitis herpetiformis	Cases	7791	49.6 ± 21.3	56.219%	64.934%	73.084%
	Controls	7791	49.6 ± 21.3	56.219%	64.934%	73.084%
Dermatitis herpetiformis *	Cases	6926	47.6 ± 21.4	58.88%	67.196%	69.448%
	Controls	6926	47.6 ± 21.4	58.88%	67.196%	69.448%
Lichen planus pemphigoides	Cases	158	56.6 ± 19.2	63.291%	68.987%	50%
	Controls	158	56.6 ± 19.2	60.127%	70.253%	50.633%
Pemphigus vulgaris	Cases	6243	56.2 ± 18.2	57.168%	60.628%	56.159%
	Controls	6243	56.2 ± 18.2	57.168%	60.628%	56.159%
Pemphigus vulgaris *	Cases	5685	53.8 ± 18.1	58.54%	60.088%	51.645%
	Controls	5685	53.8 ± 18.1	58.54%	60.088%	51.645%
Pemphigus foliaceous	Cases	4161	55.4 ± 19.1	55.756%	62.437%	56.381%
	Controls	4161	55.4 ± 19.1	55.756%	62.437%	56.381%
Paraneoplastic pemphigus	Cases	176	59.1 ± 18.1	53.977%	69.886%	68.75%
	Controls	176	58.9 ± 18.4	52.273%	72.159%	68.75%

**Table 3 biomolecules-13-00485-t003:** Pruritus is associated with a higher risk for the subsequent development of pemphigus and pemphigoid diseases. The risk of developing any of the listed pemphigus or pemphigoid diseases for individuals diagnosed with pruritus (cases, ICD10:L29) compared to those without (controls, ICD10:Z00, ***not*** ICD10:L29). Data shown display results from measures of association, excluding patients with the corresponding pemphigus or pemphigoid disease prior to the time window for the hazard ratio. Kaplan-Meier analysis (again excluding patients with outcome prior to the time window) with log-rank test was performed. Abbreviations: N: number, n.s.: not significant, n.a.: not applicable. Pemphigus or pemphigoid diseases where presence of pruritus increases the risk are highlighted in bold. A decreased risk of any pemphigus or pemphigoid disease subsequent to the diagnosis of pruritus is highlighted in blue and bold. Please note that propensity score matching is re-run with analysis of each outcome so that the analysis uses the most current data available on the TriNetX network. Analysis was performed on 11 October 2022. * Cases and controls with elevated bilirubin or urea nitrogen concentrations in serum, plasma, or blood were excluded from analysis. ** Cases and controls with any other dermatosis associated with itch were excluded from analysis.

Disease	ICD10 Code	N of Eligible Participants	N of Outcomes	Risk, %	N of Eligible Participants *	N of Outcomes	Risk, %	Risk Difference (95% Confidence Interval), %	Hazard Ratio (95% Confidence Interval)	*p* Value
Bullous pemphigoid	L12.0	1,717,744	2003	0.117	1,719,818	300	0.017	0.099% (0.094%, 0.105%)	**6.952** (6.158, 7.85)	<0.0001
Bullous pemphigoid *	L12.0	1,656,086	1561	0.094%	1,657,705	238	0.014%	0.08% (0.075%, 0.085%)	**6.794** (5.927, 7.788)	<0.0001
Bullous pemphigoid **	L12.0	1,311,489	747	0.057%	1,312,478	159	0.012%	0.045% (0.04%, 0.049%)	**4.878** (4.11, 5.789)	<0.0001
Mucous membrane pemphigoid	L12.1	1,719,520	256	0.015%	1,719,972	65	0.004%	0.011% (0.009%, 0.013%)	**4.219** (3.213, 5.54)	<0.0001
Epidermolysis bullosa acquisita	L12.3	1,719,859	180	0.01%	1,720,024	35	0.002%	0.008% (0.007%, 0.01%)	**5.402** (3.76, 7.76)	<0.0001
Dermatitis herpetiformis	L13.0	1,718,888	843	0.049%	1,719,881	146	0.008%	0.041% (0.037%, 0.044%)	**6.068** (5.09, 7.234)	<0.0001
Dermatitis herpetiformis *	L13.0	1,656,838	803	0.048%	1,657,717	152	0.009%	0.039% (0.036%, 0.043%)	**5.545** (4.662, 6.596)	<0.0001
Dermatitis herpetiformis **	L13.0	1,312,162	284	0.022%	1,312,520	85	0.006%	0.015% (0.012%, 0.018%)	**3.509** (2.753, 4.472)	<0.0001
Lichen planus pemphigoides	L43.1	1,720,023	27	0.002%	1,720,045	10	0.001%	0.001% (0%, 0.002%)	**28.705** (3.9, 211.299)	<0.0001
Pemphigus vulgaris	L10.0 or L10.1	1,719,203	522	0.03%	1,719,927	119	0.007%	0.023% (0.021%, 0.026%)	**4.582** (3.754, 5.592)	<0.0001
Pemphigus vulgaris *	L10.0 or L10.1	1,657,121	404	0.024%	1,657,758	89	0.005%	0.019% (0.016%, 0.022%)	**4.749** (3.775, 5.976)	<0.0001
Pemphigus vulgaris **	L10.0 or L10.1	1,312,179	244	0.019%	1,312,504	66	0.005%	0.014% (0.011%, 0.016%)	**3.857** (2.938, 5.063)	<0.0001
Pemphigus foliaceous	L10.2 or L10.4	1,719,417	343	0.02%	1,719,955	70	0.004%	0.016% (0.014%, 0.018%)	**5.093** (3.938, 6.587)	<0.0001
Paraneoplastic pemphigus	L10.81	1,720,022	34	0.002%	1,720,040	10	0.001%	0.001% (0.001%, 0.002%)	**6.913** (2.703, 17.678)	<0.0001

**Table 4 biomolecules-13-00485-t004:** Pruritus is associated with a higher risk for the development of pemphigus and pemphigoid diseases (6 months follow-up). The risk of developing any of the listed pemphigus or pemphigoid diseases individuals diagnosed with pruritus (cases, ICD10:L29) compared to those without (controls, ICD10:Z00, ***not*** ICD10:L29) within six months after the diagnosis of pruritus. Data shown display results from measures of association, excluding patients with the corresponding pemphigus or pemphigoid disease prior to the time window for the hazard ratio. Kaplan-Meier analysis (again excluding patients with outcome prior to the time window) with log-rank test was performed. Abbreviations: N: number, n.s.: not significant, n.a.: not applicable. Pemphigus or pemphigoid diseases where presence of pruritus increases the risk are highlighted in bold. Please note that propensity score matching is re-run with analysis of each outcome so that the analysis uses the most current data available on the TriNetX network. Analysis was performed on 11 October 2022. * Cases and controls with elevated bilirubin or urea nitrogen concentrations in serum, plasma, or blood were excluded from analysis.

Disease	ICD10 Code	N of Eligible Participants	N of Outcomes	Risk, %	N of Eligible Participants *	N of Outcomes	Risk, %	Risk Difference (95% Confidence Interval), %	Hazard Ratio (95% Confidence Interval)	*p* Value
Bullous pemphigoid	L12.0	1,717,744	973	0.057	1,719,818	41	0.002	0.054 (0.051, 0.058)	**23.293** (17.042, 31.838)	<0.0001
Mucous membrane pemphigoid	L12.1	1,719,520	79	0.005%	1,719,972	12	0.001%	0.004% (0.003%, 0.005%)	**6.459** (3.519, 11.855)	<0.0001
Epidermolysis bullosa acquisita	L12.3	1,719,859	65	0.004%	1,720,024	10	0.001%	0.003% (0.002%, 0.004%)	**12.756** (5.137, 31.679)	<0.0001
Dermatitis herpetiformis	L13.0	1,718,888	341	0.02%	1,719,881	23	0.001%	0.019% (0.016%, 0.021%)	**14.535** (9.529, 22.171)	<0.0001
Lichen planus pemphigoides	L43.1	1,718,397	10	0.001%	1,718,416	0	0%	0.001% (0%, 0.001%)	**n.a.**	0.0030
Pemphigus vulgaris	L10.0 or L10.1	1,717,578	246	0.014%	1,718,294	14	0.001%	0.014% (0.012%, 0.015%)	**17.247** (10.066, 29.553)	<0.0001
Pemphigus foliaceous	L10.2 or L10.4	1,717,792	179	0.01%	1,718,326	10	0.001%	0.01% (0.008%, 0.011%)	**29.265** (12.974, 66.013)	<0.0001
Paraneoplastic pemphigus	L10.81	1,718,397	21	0.001%	1,718,419	0	0%	0.001% (0.001%, 0.002)	**n.a.**	<0.0001

**Table 5 biomolecules-13-00485-t005:** Pemphigus and pemphigoid diseases are associated with an increased risk for pruritus. The risk probability of pruritus in individuals diagnosed with any pemphigus or pemphigoid disease (cases) compared to those without the diagnosis of any pemphigus or pemphigoid disease (controls) within 6 months after the diagnosis of any pemphigus or pemphigoid disease. Data shown display results from measures of association, excluding patients with outcome prior to the time window for the hazard ratio. Kaplan-Meier analysis (again excluding patients with outcome prior to the time window) with log-rank test was performed. Abbreviations: N: number, n.s.: not significant, n.a.: not applicable. Pemphigus and pemphigoid diseases with subsequent risk of pruritus are highlighted in bold. Pemphigus and pemphigoid diseases without subsequent risk of pruritus are highlighted in blue and bold. Pemphigus and pemphigoid diseases in which pruritus does not occur differently are displayed in grey letters, and risk difference and hazard ratio are indicated as “-“. Please note that propensity score matching is re-run with analysis of each outcome so that the analysis uses the most current data available on the TriNetX network. Analysis was performed from 10 October 2022. * Cases and controls with elevated bilirubin or urea nitrogen concentrations in serum, plasma, or blood were excluded from analysis.

Disease	ICD10 Code	N of Eligible Participants	N of Outcomes	Risk, %	N of Eligible Participants *	N of Outcomes	Risk, %	Risk Difference (95% Confidence Interval), %	Hazard Ratio (95% Confidence Interval)	*p* Value
Bullous pemphigoid	L12.0	13,740	467	3.399	15,608	110	0.705	2.694 (2.364, 3.02%)	**5.084** (4.13, 6.257)	< 0.0001
Bullous pemphigoid *	L12.0	11,643	346	2.972%	13,048	79	0.605%	2.366% (2.03%, 2.702%)	**5.093** (3.989, 6.504)	< 0.0001
Mucous membrane pemphigoid	L12.1	3888	79	2.032%	4,053	26	0.642%	1.39% (0.883%, 1.897%)	**3.223** (2.069, 5.02)	<0.0001
Epidermolysis bullosa acquisita	L12.3	1031	32	3.104%	1,199	10	0.834%	2.27% (1.093%, 3.447%)	**6.018** (2.516, 14.393)	<0.0001
Dermatitis herpetiformis	L13.0	6532	197	3.016%	7,638	37	0.484%	2.532% (2.088%, 2.975%)	**6.263** (4.409, 8.899)	<0.0001
Dermatitis herpetiformis *	L13.0	5731	175	3.054%	6,761	37	0.547%	2.506% (2.027%, 2.985%)	**5.632** (3.951, 8.03)	<0.0001
Lichen planus pemphigoides	L43.1	127	10	7.874%	154	10	6.494%	1.381% (−4.71%, 7.471%)	**7.042** (0.848, 58.497)	0.0350
Pemphigus vulgaris	L10.0 or L10.1	5803	115	1.982%	6,109	29	0.475%	1.507% (1.109%, 1.905%)	**4.198** (2.793, 6.308)	<0.0001
Pemphigus vulgaris *	L10.0 or L10.1	5334	87	1.631%	5,544	22	0.397%	1.234% (0.856%, 1.612%)	**4.043** (2.533, 6.454)	<0.0001
Pemphigus foliaceous	L10.2 or L10.4	3895	84	2.157%	4,068	25	0.615%	1.542% (1.027%, 2.058%)	3.471 (2.221, 5.424)	<0.0001
Paraneoplastic pemphigus	L10.81	146	10	6.849%	169	10	5.917%	-	-	n.s.

## Data Availability

All data presented in this work are shown in the tables.

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
