# Peer review of "Pruritus Is Associated with an Increased Risk for the Diagnosis of Autoimmune Skin Blistering Diseases: A Propensity-Matched Global Study"

_biomolecules, 2023, doi:10.3390/biom13030485_

Round 1
Reviewer 1 Report
This field study is a study of great interest that demonstrates the relationship between the appearance of pruritus and the loss of quality of life in various pathologies related to this symptom, but after a study of a large number of patients and controls.
Since it is a significant sample, it is concluded that the presence of pruritus in AIBP should be taken into account when diagnosing patients, an autoimmune disease that is often underdiagnosed.
The reference to the different antigenic markers related to pruritus is in turn of great interest, and, as an expert in the field, I would like to provide you with a piece of information: both the TRPV channels and the TRPA channels, which the bibliography indicates are related to pruritus. , are also nociceptive channels, hence the relevance of pruritus and these molecules, since it goes from being an annoying symptom to activating nociceptive neurons and, therefore, the pain pathways.
Thank you once again for counting on me for this review. A great job.
Author Response
This field study is a study of great interest that demonstrates the relationship between the appearance of pruritus and the loss of quality of life in various pathologies related to this symptom, but after a study of a large number of patients and controls. (…) Thank you once again for counting on me for this review. A great job.
Thank you for your very positive feedback and the mentioning of the TRPV channels and the TRPA channels.
Reviewer 2 Report
The article is very interesting and well prepared. It adds a lot to the knowledge of autoimmune diseases and itching. However, one should remember about other possible causes of pruritus in this group of patients, e.g. cholestasis or uremia.
Author Response
The article is very interesting and well prepared. It adds a lot to the knowledge of autoimmune diseases and itching. However, one should remember about other possible causes of pruritus in this group of patients, e.g. cholestasis or uremia.
Thank you very much for this important point. Prompted by your comment, we investigated the number of EBA patients diagnosed with “Unspecified kidney failure” (ICD10:N19, n=10) and “Toxic liver disease with cholestasis” (ICD10:K71.0, n=0). Because of these small numbers, we refrained to perform propensity matching based on these parameters. It is also not possible to account for values of laboratory measurements as outcomes. However, we performed an analysis, where we excluded cases and controls with elevated total bilirubin (>1.2 mg/dl) or with elevated urea nitrogen (>24 mg/dl) for the 3 AIBDs with the highest patient numbers (BP, DH and PV) and re-run the same analysis. In materials and methods we state:
“To exclude that presence of cholestasis or uremia confound the results, we performed a subgroup analysis where we excluded cases and controls (for all groups) that had elevated bilirubin concentration in serum, plasma, or blood (TNX:9050, most recent laboratory, >1.2 mg/dl) or elevated urea nitrogen in serum, plasma, or blood (TNX:9030, most recent laboratory, >24 mg/dl). These analyses were done on the Global Collaborative Network on January 25th, 2023.”
In the results section we state:
“To exclude confounding by presence of cholestasis or uremia, subgroup analysis was performed for the 3 most prevalent AIBD within the TriNetX network (BP, DH, and PV) excluding cases and controls with elevated bilirubin or urea nitrogen concentrations in serum, plasma, or blood. As expected, cohort sizes decreased. As for the cohorts prior to consideration of these laboratory results, no significant differences were noted among the factors after propensity-matching (table 1). In this subgroup analysis, the risk for developing BP, DH or PV following pruritus was comparable to the analysis that did not exclude elevated bilirubin or urea nitrogen concentrations (table 3). This indicates that the development of BP, DH of PV following pruritus is mainly independent of cholestasis or uremia.”
and
“To exclude confounding by presence of cholestasis or uremia, subgroup analysis was performed for the 3 most prevalent AIBD within the TriNetX network (BP, DH and PV) excluding cases and controls with elevated bilirubin or urea nitrogen concentrations in serum, plasma, or blood. As expected, cohort sizes decreased. As for the cohorts prior to consideration of these laboratory results, no significant differences were noted among the factors after propensity-matching (table 2). In this subgroup analysis, the risk for developing pruritus following either BP, DH, or PV, was comparable to the analysis that did not exclude elevated bilirubin or urea nitrogen concentrations (table 5). This indicates that the development of pruritus in these AIBDs is mainly independent of cholestasis or uremia.”
The respective newly generated data has been added to the tables.
Reviewer 3 Report
Dear authors,
I read your manuscript with great interest. Well-written and easy to follow. My comments are as below:
- In the introduction part citations are missing in the sedond paragraph. This must be corrected.
- In the introduction part as well as figure legends terms used for the first time should be mentioned complete and not with abbreviations.
- Page 3, line 108: "quality of life" and not "quality of live"
- How the access to TrinetX was provided? do authors needed to pay for these data? Trintex is a private company. This must be clearly explained in the manuscript. From which countries are these data collected?
- In table 1: the average age of patients is 43.3 +/- 23.8. The BP patients are generally older patients. Does this age category covers potential BP patients?
Results and conclusion parts are clear.
Author Response
I read your manuscript with great interest. Well-written and easy to follow. My comments are as below:
- In the introduction part citations are missing in the second paragraph. This must be corrected.
Thank you, we added additional references as suggested.
- In the introduction part as well as figure legends terms used for the first time should be mentioned complete and not with abbreviations.
Thank you. We implemented your suggestion.
- Page 3, line 108: "quality of life" and not "quality of live"
Changed as suggested.
- How the access to TrinetX was provided? do authors needed to pay for these data? Trintex is a private company. This must be clearly explained in the manuscript. From which countries are these data collected?
This important information has been added to the Materials and Methods section:
- In table 1: the average age of patients is 43.3 +/- 23.8. The BP patients are generally older patients. Does this age category covers potential BP patients?
Table 1 displays cases and controls for pruritus. The demographics for BP (and all other AIBD) patients are shown in table 2. The average age of the BP patients (not considering bilirubin or urea nitrogen concentrations) is 72.7 ± 14.9 years.
Reviewer 4 Report
BP, EBA, LABD and DH are diseases in which pruritus is a constant symptom and a typical feature. It has long been known that in these diseases pruritus may precede the appearance of active skin lesions. The general consensus is that in any case of prolonged pruritus without skin lesions or pruritus with atypical skin lesions, direct and indirect immunopathology should be performed to exclude autoimmune bullous disease. Pemphigus is a potentially life-threatening AIBD and characterized by flaccid fragile blisters and erosions of the skin and/or mucous membranes. In contrast to BP, pruritus is less frequently present and with lower intensity in the pemphigus group.
According to Ghodsi et al. the most common subjective symptoms reported by patients with pemphigus vulgaris are burning (83.1%), pain (68.4%) and pruritus (47.5%). Pemphigus foliaceus is another disease of this group. Pemphigus foliaceus is another disease from this group. Here, pruritus occurs in more than half of the patients (61%). The inflammation might be of great relevance for the induction of pruritus.
Therefore, I believe that the assumption of the work is known and does not add anything new to the knowledge about these diseases. The second problem is, in my opinion, an incorrect assumption to take into account only diagnoses from ICD 10 L29, Z00. The problem of misdiagnosis of AIBD is also the diagnosis with the ICD 10 L50. The authors themselves state that the work is limited to a small group and includes performance in one academic center and the lack of control of coexisting pruritic conditions.
And the authors themselves admit that that relative low number of BP patients with pruritus may also be due to the study design. And the authors themselves admit that this relatively small number of BP patients with pruritus may also be due to the design of the study. But in my opinion there is the same problem with DH. It is known to be a very itchy disease that goes undiagnosed for many months.
The authors give numerous self-citations (about different disorders), but do not list several works that would be suitable for publication. For example:
· Meijer JM, Lamberts A, Luijendijk HJ, et al. Prevalence of Pemphigoid as a Potentially Unrecognized Cause of Pruritus in Nursing Home Residents. JAMA Dermatol. 2019;155(12):1423–1424.
· Lilianna Kulczycka-Siennicka, Anna Cynkier, Elżbieta Waszczykowska, Anna Woźniacka, Agnieszka Å»ebrowska, "The Role of Intereukin-31 in Pathogenesis of Itch and Its Intensity in a Course of Bullous Pemphigoid and Dermatitis Herpetiformis", BioMed Research International, vol. 2017, Article ID 5965492, 8 pages, 2017.
· Kanwar A J, Gupta R, Kaur S. Pruritus - A Clinical Sign of Activity of Pemphigus. Indian J Dermatol Venereol Leprol 1989;55:396
· Cole, Emily F. MD, MPHa; DeGrazia, Taryn MD; AlShamekh, Shomoukh MDb; Feldman, Ron MD, PhDa,. Itch-related quality of life impact across 3 autoimmune blistering diseases: a retrospective cohort study. Itch 5(3):p e39, July-September 2020
· Papara C, Danescu S, Sitaru C, Baican A. Challenges and pitfalls between lichen planus pemphigoides and bullous lichen planus. Australas J Dermatol. 2022 May;63(2):165-171. doi: 10.1111/ajd.13808. Epub 2022 Feb 23. PMID: 35196400.
· Ujiie H, Yamagami J, Takahashi H, Izumi K, Iwata H, Wang G, Sawamura D, Amagai M, Zillikens D. The pathogeneses of pemphigus and pemphigoid diseases. J Dermatol Sci. 2021 Dec;104(3):154-163. doi: 10.1016/j.jdermsci.2021.11.003. PMID: 34916040.
The pathogenesis of pruritus in each of these diseases has a slightly different background. And not only the nervous system or Il-31 mentioned by the authors are important, but many other factors, including mast cell mediators, neuropeptides. The discussion does not refer at all to publications on DH, EBA or LPP, although the authors have studied such diseases.
Author Response
BP, EBA, LABD and DH are diseases in which pruritus is a constant symptom and a typical feature. It has long been known that in these diseases pruritus may precede the appearance of active skin lesions. The general consensus is that in any case of prolonged pruritus without skin lesions or pruritus with atypical skin lesions, direct and indirect immunopathology should be performed to exclude autoimmune bullous disease.
We are in full agreement with the reviewer. This general consensus, has so far, however been based on expert opinion. To the best of our knowledge, this had so far not been addressed in any systematic epidemiological investigation. The data provided herein now provide solid evidence for this statement.
Pemphigus is a potentially life-threatening AIBD and characterized by flaccid fragile blisters and erosions of the skin and/or mucous membranes. In contrast to BP, pruritus is less frequently present and with lower intensity in the pemphigus group. According to Ghodsi et al. the most common subjective symptoms reported by patients with pemphigus vulgaris are burning (83.1%), pain (68.4%) and pruritus (47.5%). Pemphigus foliaceus is another disease of this group. Pemphigus foliaceus is another disease from this group. Here, pruritus occurs in more than half of the patients (61%). The inflammation might be of great relevance for the induction of pruritus.
We apologize for not considering the work of Ghodsi et al. (PMID: 21967321), which we have now included in the manuscript. We agree with the reviewer that inflammation is the most likely driver of itch. This, however, cannot be derived from the data presented herein. Regarding the statement that “In contrast to BP, pruritus is less frequently present (in pemphigus)”, data published by Rolader et al. (PMID: 31962093) clearly documents that prevalence and severity are identical in pemphigus and BP patients.
Therefore, I believe that the assumption of the work is known and does not add anything new to the knowledge about these diseases.
Here we kindly disagree with the reviewer, because (i) a temporal relationship between pruritus and AIBD has so far not been investigated, (ii) pruritis in AIBDs other than pemphigus and BP had never been investigated, (iii) data from propensity-matched investigations on that subject had also not been presented to the best of out knowledge. As our article addresses all 3 aspects, the data presented is novel and adds to new insights into AIBD.
The second problem is, in my opinion, an incorrect assumption to take into account only diagnoses from ICD 10 L29, Z00.
This raises an important point. ICD10:Z00 includes “Encounter for general examination without complaint, suspected or reported diagnosis” and identifies individuals in contact with the health care system with the least probability of an additional diagnosis. ICD10:L29 codes for all patients with a diagnosis of pruritus and identifies all patients with this specific condition. To ensure that other diseases that are associated with pruritus did not confound the results, we performed a subgroup analysis for the 3 most prevalent AIBDs (BP, DH and PV) in out cohort. Here we compared controls (ICD10:Z00) to cases (ICD10:L29) in the absence of urticaria (ICD10:L50), psoriasis (ICD10:L40) and dermatitis and eczema (ICD10:L20-L28 and L30). This subgroup analysis has been included in the revised version of the manuscript.
The problem of misdiagnosis of AIBD is also the diagnosis with the ICD 10 L50.
We are sorry, but we do not fully comprehend this remark ICD10:L50 refers to urticaria. We do not understand how this relates to the data presented herein.
The authors themselves state that the work is limited to a small group and includes performance in one academic center and the lack of control of coexisting pruritic conditions.
We agree that our study lacked control of coexisting pruritic conditions and thank the reviewer for this valuable comment. To address this this, we add a new subgroup analysis and added the following to the revised version of the manuscript:
Materials and Methods:
To exclude that presence of other inflammatory skin diseases in which pruritus is a common symptom is a confounder, we performed a subgroup analysis comparing cases (pruritus, ICD10:L29) to controls (ICD10:Z00) excluding (for both groups) ICD10 codes for urticaria (ICD10:L50), psoriasis (ICD10:L40) and dermatitis and eczema (ICD10:L20-L28 and L30) present at any until the index event (ICD10:L29 or ICD10:Z00). Both subgroup analyses were performed on the Global Collaborative Network on January 25th, 2023.
Results:
To exclude that presence of other inflammatory skin diseases in which pruritus is a common symptom is a confounder, we performed a subgroup analysis excluding ICD10 codes for inflammatory sin diseases in which pruritus is commonly observed from both groups. The cohorts showed no difference regarding age, sex, ethnicity, and skin color (table 1). The risk of developing BP, DH or PV following the diagnosis of pruritus was also significantly increased when aforementioned inflammatory skin diseases were excluded. However, compared to the initial analysis and the subgroup analysis considering elevated bilirubin or urea nitrogen concentrations, the hazard rations were lower (table 3). This indicates that pruritus is associated with an increased risk for BP, DH, and PV in-dependent of the presence of other inflammatory skin diseases.
Tables 1 and 3 were adopted accordingly, including this newly generated analysis.
We are again sorry, but we do not fully understand this the remaining aspects of the comment. The limitation of a small group is mentioned once in the manuscript and refers to the small number of BP patients additionally diagnosed with ICD10:L29. The wording in the “Limitations” section of the manuscript has been reworded to better reflect the indented meaning (see our answer to your question below). Other than that, we never mentioned that a small sample size is a potential limitation of our study. We also believe that the sample sizes for ICD10:L29 of over 1.5 million patients are more than sufficient for our investigation. In line, given the prevalence of AIBDs, sample sizes for the AIBDs investigated are in our opinion also suited to address the herein mentioned research questions. We also do not understand your statement “(study) includes performance in one academic center” because (as stated in the materials and methods section) the study was performed in the Global Collaborative Network of TriNetX, which at the time of analysis had 91 contributing health care organizations.
And the authors themselves admit that that relative low number of BP patients with pruritus may also be due to the study design. And the authors themselves admit that this relatively small number of BP patients with pruritus may also be due to the design of the study. But in my opinion there is the same problem with DH. It is known to be a very itchy disease that goes undiagnosed for many months.
We are in full agreement with the reviewer. With the statement “Third, coding of a symptom such as pruritus, may be biased. More specifically, pruritus may not be coded alongside a disease that is associated with itch – such as BP. In our dataset this may be the case, as “only” 3.4% of BP patients developed pruritus. This relative low number of BP patients with pruritus may also be due to the study design, as outcomes prior to the index event were excluded.” We intended to point out that when a definite other diagnosis that is associated with pruritus, e.g., BP, has been established, coding for pruritus is most likely underrepresented. We have changed the text in the revised manuscript accordingly. We also agree that similar considerations as mentioned for BP apply to DH. We, however, wish to refrain to discuss this in detail, as the reference to BP is meant as an example for all AIBDs included here.
The authors give numerous self-citations (about different disorders), ….
Following your suggestion, we deleted the following sentence and references: “We performed global population-based studies with a propensity-matched case-control design following established protocols [20–23].”, thereby reducing the number of self-citations by 4.
….but do not list several works that would be suitable for publication. For example:
- Meijer JM, Lamberts A, Luijendijk HJ, et al. Prevalence of Pemphigoid as a Potentially Unrecognized Cause of Pruritus in Nursing Home Residents. JAMA Dermatol. 2019;155(12):1423–1424.
We had already included this important paper in the reference list (please see reference 16 in the R0 version).
- Lilianna Kulczycka-Siennicka, Anna Cynkier, Elżbieta Waszczykowska, Anna Woźniacka, Agnieszka Żebrowska, "The Role of Intereukin-31 in Pathogenesis of Itch and Its Intensity in a Course of Bullous Pemphigoid and Dermatitis Herpetiformis", BioMed Research International, vol. 2017, Article ID 5965492, 8 pages, 2017.
Thank you, we have included this reference.
- Kanwar A J, Gupta R, Kaur S. Pruritus - A Clinical Sign of Activity of Pemphigus. Indian J Dermatol Venereol Leprol 1989;55:396
We wish to refrain to include this reference because it does not provide any data, but rather states the opinion of the authors that is based on their clinical experience
- Cole, Emily F. MD, MPHa; DeGrazia, Taryn MD; AlShamekh, Shomoukh MDb; Feldman, Ron MD, PhDa,. Itch-related quality of life impact across 3 autoimmune blistering diseases: a retrospective cohort study. Itch 5(3):p e39, July-September 2020
Thank you, we have included this reference.
- Papara C, Danescu S, Sitaru C, Baican A. Challenges and pitfalls between lichen planus pemphigoides and bullous lichen planus. Australas J Dermatol. 2022 May;63(2):165-171. doi: 10.1111/ajd.13808. Epub 2022 Feb 23. PMID: 35196400.
We would like to refrain to cite this review.
- Ujiie H, Yamagami J, Takahashi H, Izumi K, Iwata H, Wang G, Sawamura D, Amagai M, Zillikens D. The pathogeneses of pemphigus and pemphigoid diseases. J Dermatol Sci. 2021 Dec;104(3):154-163. doi: 10.1016/j.jdermsci.2021.11.003. PMID: 34916040.
We would like to refrain to cite this review.
The pathogenesis of pruritus in each of these diseases has a slightly different background. And not only the nervous system or Il-31 mentioned by the authors are important, but many other factors, including mast cell mediators, neuropeptides. The discussion does not refer at all to publications on DH, EBA or LPP, although the authors have studied such diseases.
Thanks for mentioning that we have published on DH, EBA or LPP, which are the topic of other publications in our group. Here, we focus on itch and AIBD, thus taking the important comment of the reviewer into account to expand the discussion accordingly:
Beside IL-31 thymic stromal lymphopoietin (TSLP) is a cytokine described as key mediator in pruritus, which can activate neurons to induce itch [39]. TSLP is involved in the pathogenesis of BP [40]. A recent study identified by RNA sequencing itch-related me-diators and receptors, such as phospholipase, substance P, sodium channels, TRP channels and different cytokines and chemokines, that are differentially expressed in pruritic skin [41]. It has been found that many other factors, including mast cell mediators, neuro-peptides, neurotrophins and proteases have many effects on the skin, mainly participating in the occurrence and development of itching [42]. Although the exact mechanism of itch has not been fully elucidated, it is clear that a complex crosstalk between the stratum corneum, keratinocytes, T cells, eosinophils, basophils, mast cells and nerve fibers play an important role in the initiation and maintenance of pruritus [43–45].
Round 2
Reviewer 4 Report
I accept in present form